# Environmental Pollutants Impair Transcriptional Regulation of the Vitellogenin Gene in the Burrowing Mud Crab (*Macrophthalmus Japonicus*)

**Kiyun Park [1], Hyunbin Jo [1], Dong-Kyun Kim [1]** 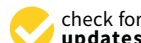 **and Ihn-Sil Kwak [1,2,*]**

[1] Fisheries Science Institute, Chonnam National University, Yeosu 59626, Korea; ecoblue@hotmail.com (K.P.); prozeva@hanmail.net (H.J.); dkkim1004@gmail.com (D.-K.K.)
[2] Faculty of Marine Technology, Chonnam National University, Yeosu 550-749, Korea
* Correspondence: iskwak@chonnam.ac.kr; Tel.: +82-61-6597148; Fax: +82-61-6597149

**Abstract:** Vitellogenesis is a pivotal reproductive process of the yolk formation in crustaceans. Vitellogenin (VTG) is the precursor of main yolk proteins and synthesized by endogenous estrogens. The intertidal mud crab (*Macrophthalmus japonicus*) inhabits sediment and is a good indicator for assessing polluted benthic environments. The purpose of this study was to identify potential responses of *M. japonicus* VTG under environmental stresses caused by chemical pollutants, such as 1, 10, and 30 μg L$^{-1}$ concentrations in di(2-ethylhexyl) phthalate (DEHP), bisphenol A (BPA) and irgarol. We characterized the *M. japonicus* VTG gene and analyzed the transcriptional expression of VTG mRNA in *M. japonicus* exposed to various chemicals and exposure periods. A phylogenetic analysis revealed that the *M. japonicus* VTG clustered closely with *Eriocheir sinensis* (Chinese mitten crab) VTG, in contrast with another clade that included the VTG ortholog of other crabs. The basal level of VTG expression was the highest in the hepatopancreas and ovaries, and tissues. VTG expression significantly increased in the ovaries and hepatopancreas after 24 h exposure to DEHP. Increased responses of VTG transcripts were found in *M. japonicus* exposed to DEHP and BPA for 96 h; however, VTG expression decreased in both tissues after irgarol exposure. After an exposure of 7 d, VTG expression significantly increased in the ovaries and hepatopancreas for all concentrations of all chemicals. These results suggest that the crustacean embryogenesis and endocrine processes are impaired by the environmental chemical pollutants DEHP, BPA, and irgarol.

**Keywords:** vitellogenin (VTG); crustacean; di(2-ethylhexyl) phthalate (DEHP); bisphenol A (BPA); irgarol

## 1. Introduction

Vitellogenesis is a hormonally regulated process for synthesizing yolk proteins and an important step in the reproductive development of crustaceans [1]. Vitellogenin (VTG), a precursor molecule of vitellin, is stored in crustacean hepatopancreas tissue and transported by the hemolymph to the ovaries. The sites of VTG synthesis are in the hepatopancreas and ovary tissues in decapod crustaceans [2–5]. VTG genes have shown specific levels of expressions that vary by tissue, sex, and development stage [6]. VTG inductions were observed in aquatic invertebrates by exposure to estrogenic compounds [7,8]. After endosulfan exposure, VTG gene expression was downregulated in Pandalus shrimp (*Pandalopsis japonica*) [9]. The regulation of VTG gene may be affected by several pollutants [7–9].

Environmental chemical pollution is globally one of the most critical ecological problems, owing to the high risks it poses to ecosystems and human life [10]. Bisphenol A (BPA), a common endocrine

disrupting chemical (EDC), is a carbon-based synthetic compound and a xenoestrogen that interferes with estrogen receptor signaling [11]. BPA concentrations observed in U.S. river estuaries were approximately 4 µg $L^{-1}$ to 21 µg $L^{-1}$ [12,13]. Exposure to BPA induced transcription responses of the heat shock protein 90 (HSP90) gene in the marine crab *Charybdis japonica* [14]. Di-(2-ethylhexyl) phthalate (DEHP), a suspected EDC, is widely used as plasticizer and may induce reproductive and developmental toxicities in aquatic environments [15]. DEHP toxicity induced changes in the VTG levels in African sharptooth catfish (*Clarias gariepinus*) and the Chinese rare minnow (*Gobiocypris rarus*) [16,17]. DEHP levels were 0.47 to 12 µg $L^{-1}$ in the China river estuary [18]. Irgarol, which is used as an algicide, has been found in coastal and estuarine environments because it is commonly used in antifouling systems [19,20]. Irgarol exposure induced downregulation of HSP90 expression in hard coral (*Acropora tenuis*) [21]. Worldwide, the environmental concentrations of irgarol in antifouling systems range from 1 ng $L^{-1}$ to 1 mg $L^{-1}$ [22]. Irgarol's long half-life poses an ecological risk in estuarine ecosystems [20]. Although these pollutants are ubiquitously distributed in aquatic environments, there is not enough information about the effects of embryogenesis in crustaceans.

Crabs make up one of the largest crustacean families. The accumulation and synthesis of VTG is vital for oocyte and embryo development in crabs [2,6,23]. A burrowing mud crab, *Macrophthalmus japonicus* (Ocypodoidea), inhabits estuarine intertidal mud flats throughout the Indo-Pacific regions of Japan and Korea [24–27]. *M. japonicus* is also a major bioturbator of tidal flats and plays a crucial role in purifying sediment [28]. The movement of *M. japonicus* may affect the distribution of macro-infauna because of the accompanying disturbance in the physical and chemical properties of the sediment [25]. Previous studies have suggested that *M. japonicus* may be a useful indicator for modeling the exposure effects of chemical toxicity in the sediment environment [26,27,29,30].

In this study, we assessed the effects of crustacean embryogenesis and endocrine process by environmental chemical pollutants such as BPA, DEHP, and irgarol using the intertidal mud crab. To accomplish this, we characterized the *M. japonicus* VTG gene and analyzed phylogenetic relationships. The transcriptional responses of VTG and the tissue distribution of VTG mRNA expression were investigated from *M. japonicus* ovaries and hepatopancreases, plus other tissues, following exposure to either BPA, DEHP, or irgarol.

## 2. Materials and Methods

### 2.1. Organisms

*M. japonicus* was prepared from marine product markets in Yeosu (Jeonnam, Korea). On average, crabs ranged from 2.5 to 3.5 cm in carapace height, 2.7 to 4.3 cm in carapace width and 4 to 11 g in body weight. The crabs were placed in glass containers (45.7 × 35.6 × 30.5 cm) with natural seawater and aeration. All samples acclimatized for 1 d at 18 ± 1 °C in temperature, 25% salinity, and a 12 h light-dark schedule. Non-damaged crabs were selected. Experiments were performed based on the guidelines of the Chonnam National University Institutional Animal Care and Use Committee.

### 2.2. Exposure Experiments

Irgarol (2-(tert-butylamino)-4-(cyclo-propylamino)-6-(methyl-thio)-s-triazine) and BPA (99.9% pure) were prepared from Sigma-Aldrich (St. Louis, MO, USA). DEHP solutions were obtained from a solid compound (99%, Junsei Chemical Co. Ltd., Japan). The 10 mg $L^{-1}$ stock solutions of BPA, DEHP, and irgarol were prepared by dissolving the chemicals in 99% acetone at room temperature. For working solutions of 1, 10, and 30 µg $L^{-1}$ for each chemical; the stock solution was diluted with seawater. In solvent controls, a solvent concentration was <0.5% acetone.

For each chemical exposure, the crabs (*n* = 120) were randomly divided into four experimental groups (1, 10, and 30 µg $L^{-1}$ treatment solutions, as well as solvent controls). Ten crabs were exposed with one of three doses of BPA, DEHP, or irgarol. Treatment times were 24 h, 96 h, and 7 d. Three individuals were subjected to tissue extractions for each time interval in each chemical

treatment condition and control crab. The seawater was changed every day by adding the equivalent concentration of each chemical during the experiment. No food was provided during the experimental period. All experiments were run in three replicates with independent samples. Following exposure, tissue samples were immediately extracted from the crabs.

### 2.3. Characterization of the M. Japonicus Vitellogenin (VTG) Genes and Phylogenetic Analysis

VTG gene sequences were obtained using the 454 GS FLX transcriptomic database of the *M. japonicus* body [31]. The identified VTG cDNA sequences were compared with the VTG sequences in crustacean species available on the national center for biotechnology information (NCBI) database, using basic local alignment search tool (BLAST) searching (http://www.ncbi.nlm.nih.gov). The deduced amino acid sequences were obtained using the ExPASy translation program and aligned using the Clustal W2 tool.

To analyze the *M. japonicus* VTG gene in a phylogenetic tree, 14 crustacean VTG sequences were downloaded from NCBI for comparison of their similarities with the deduced amino acid sequences of the *M. japonicus* VTG. The GeneDoc Program (ver. 2.6.001) was used to display the multiple aligned sequences. A construction of the phylogenetic tree was made by neighbor-joining analyses of the Mega X program (version 10.04). The bootstrap value was calculated by 1000 replicates.

### 2.4. RNA Isolation and mRNA Expression Analysis

Total RNA from *M. japonicus* was obtained using the RNAiso Plus reagent (Takara, Dalian, China) following the manufacturer's protocol. Recombinant DNase I (RNase free) (Takara, Kusatsu, Japan) treatment was used to eliminate contamination of genomic DNA. Integrity and quantity of the extracted RNA were checked using 0.8% agarose gel for electrophoresis and a Nano-Drop 1000 (Thermo Fisher Scientific, Foster City, CA, USA). cDNA synthesis was performed with 3 μg of total RNA using the SuperScript™III RT kit (Invitrogen, Foster City, CA, USA) following the manufacturer's protocol. After synthesis, the diluted cDNA (50-fold) was stored at $-80\ ^\circ$C.

To evaluate VTG gene responses in multiple tissues, the total RNA was extracted from *M. japonicus* in a seawater control. Quantitative real-time PCR (polymerase chain reaction, RT-PCR) was performed using the Exicycler™96 (Bioneer, Daejeon, Korea) with the master mix (Bioneer, Daejeon, Korea). The glyceraldehyde-3-phosphate dehydrogenase (*Mj GAPDH*) gene of *M. japonicus* was used as an internal reference [16]. The primer sequences were: *Mj*_VTG forward 5′-CTTGGGCTCTCCAGTTC TTG-3′; *Mj*_VTG reverse 5′-CCACGTATGCCTCTTTTGGT-3′; *Mj*_GAPDH forward 5′-TGCTGATGCACCCATGTTTG-3′; *Mj*_GAPDH reverse 5′-AGGCCCTGGACAATCTCAA AG-3′. The size of the PCR product was 164 bp for the VTG gene and 147 bp for the GAPDH gene. The reaction mixture (20 μL) contained 6 μL of diluted cDNA, 10 μL of 2x SYBR green dye (Bioneer, Daejeon, Korea), 0.5 μL of forward and reverse primers (10 μM), and 3.0 μL of RNase-free water. The following RT-PCR cycle was used for amplification of the VTG gene: 94 $^\circ$C for 40 s, followed by 38 cycles of 94 $^\circ$C for 15 s, 53 $^\circ$C for 30 s and 72 $^\circ$C for 40 s. The Exicycler™96 real time system program (version 3.54.8) was used for the verification of the RT-PCR baseline. The calculation for relative transcript levels was used by the $2^{-\Delta\,\Delta ct}$ method [32] and normalized with GAPDH.

### 2.5. Statistical Analysis

Statistical analysis was conducted using statistical package for the social sciences (SPSS) 12.0 KO (SPSS Inc., Chicago, IL, USA). Statistical analysis for tissue distribution was performed using Dunnett's multiple range test of One-Way Analysis of Variance (ANOVA). An independent sample *t*-test was used to compare significant differences in the transcriptional expression of VTG between the ovaries and hepatopancreas under different chemical concentrations. A two-way ANOVA was conducted to assess the effects of dose and exposure to each chemical on VTG expression. Differences were statistically significant at $p < 0.05$ (*) and $p < 0.01$ (**). Data are presented as the mean $\pm$ SD.

## 3. Results

### 3.1. Characterization of the M. Japonicus VTG Gene

The partial cDNA of the *M. japonicus* VTG gene was obtained from the GS-FLX transcriptome database of the *M. japonicus* crab [31]. The *M. japonicus* VTG DNA was 3700 bp long, included an open reading frame (ORF) of 1202 amino acids, and included a lipoprotein amino acid terminal region of VTG (Figure 1). The *M. japonicus* VTG nucleotide sequence was 87% homologous with that of *Eriocheir sinensis* (KC699915). The predicted amino acid sequences of the VTG were 84%, 72%, 65%, and 63% homologous with those of *E. sinensis* (AGM75775), *Longpotamon honanense* (freshwater crab; AKI23633), *Scylla paramamosain* (green mud crab; ACO36035), and *Portunus trituberculatus* (swimming crab; AAX94762), respectively. Thus, the VTG gene of *M. japonicus* does not show a high homology with that of the other crabs (Figure 1). A phylogenetic analysis revealed two clades of VTG of the crabs (Figure 2). One clade was composed of VTGs from *M. japonicus*, *E. sinensis* and *L. honanense*. Another clade had homologous VTGs from *S. paramamosain*, *P. trituberculatus*, *Charybdis feriata* (Indo-Pacific crab; AAU93694), and *Callinectes sapidus* (blue crab; ABC41925). In addition, the black tiger shrimp (*Penaeus monodon*) (ABB89953) VTG formed another clade with the VTG of lobster (*Homarus americanus*; ABO09863), multiple prawn species, and other related shrimp species (Figure 2).

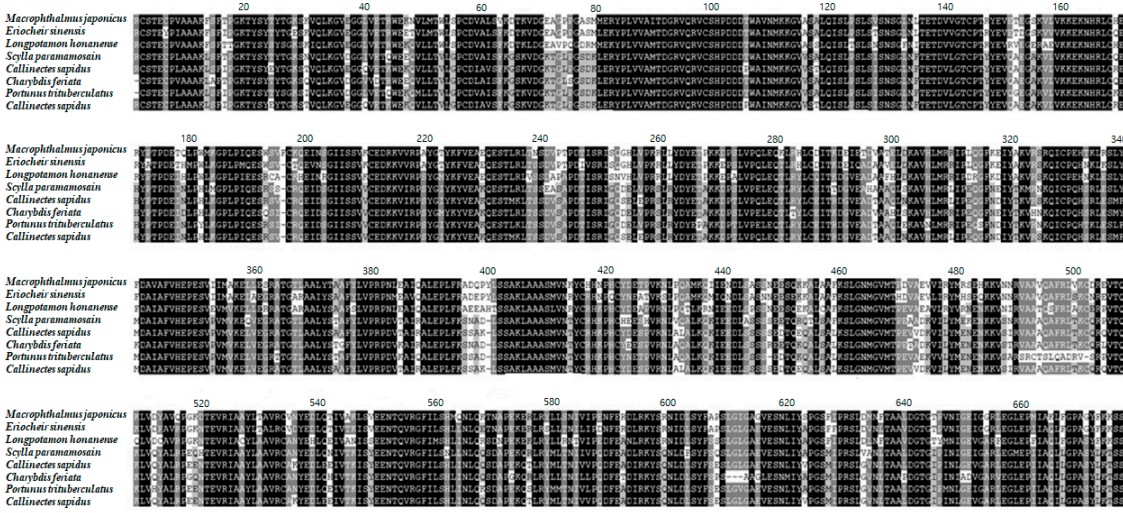

**Figure 1.** Vitellogenin (VTG) gene of the intertidal mud crab, *M. japonicus*. Multiple sequence alignments of the deduced *M. japonicus* VTG gene sequences with the homologous sequences of other crabs.

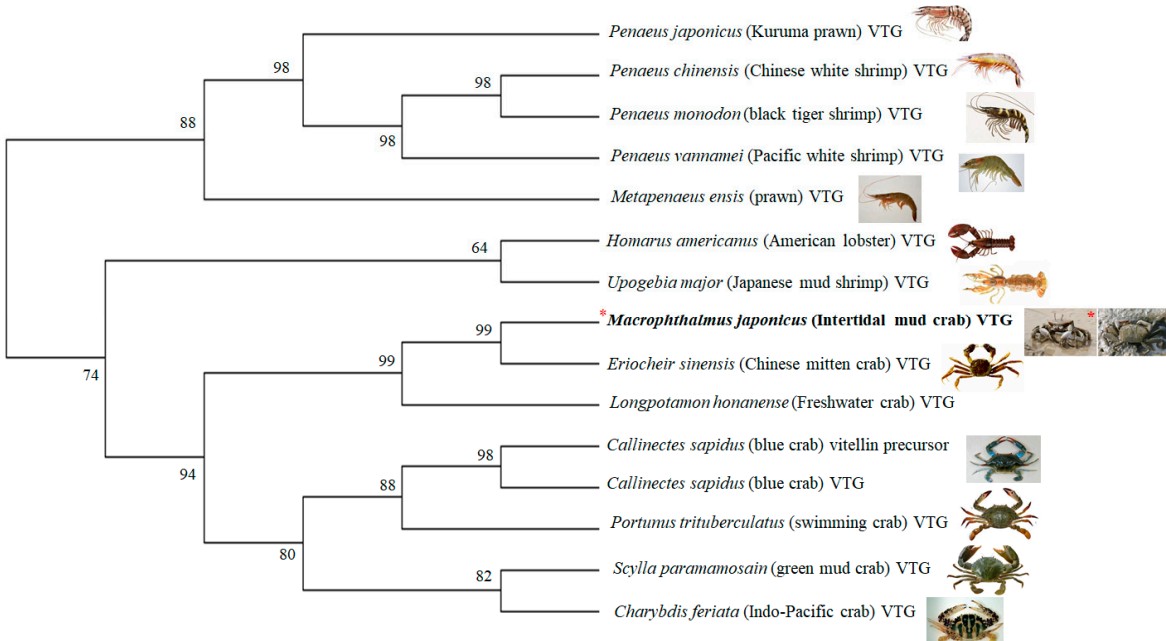

**Figure 2.** Phylogenetic tree of *M. japonicus* VTG gene constructed by neighbor-joining analysis (bootstrap value 1000). The numbers at the nodes are the percentage bootstrap values. GenBank accession numbers for VTGs are: *M. japonicus* VTG (Acc **), *Eriocheir sinensis* (KC699915), *Longpotamon honanense* (AKI23633), *Callinectes sapidus* vitellin precursor (AEI59132), *C. sapidus* VTG (ABC41925), *Portunus trituberculatus* (AAX94762), *Scylla paramamosain* (ACO36035), *Charybdis feriata* (AAU93694), *Homarus americanus* (ABO09863), *Upogebia major* (BAF91417), *Penaeus japonicus* (BAD98732), *Penaeus chinensis* (ABC86571), *Penaeus monodon* (ABB89953), *Penaeus vannamei* (AAP76571), and *Metapenaeus ensis* (AAN40700). ** Accession #s to be deposited in GenBank.

### 3.2. Basal Levels of M. Japonicus VTG Expression in Multiple Tissues

The basal expression levels of VTG were observed in six tissues (gill, hepatopancreas, muscle, ovaries, heart, and stomach) (Figure 3). The fold change of each VTG mRNA expression in each tissue was based on the VTG expression level in gill tissue (when the set value of the gill = 1). The *M. japonicus* VTG gene expression was expressed differentially in all investigated tissues. The high level of *M. japonicus* VTG expression was evident in the ovaries and hepatopancreas, whereas relatively low levels were observed in the gill, muscle, stomach, and heart. The *M. japonicus* VTG mRNA expression varied significantly between the crab gill and ovaries, or hepatopancreas ($p < 0.05$).

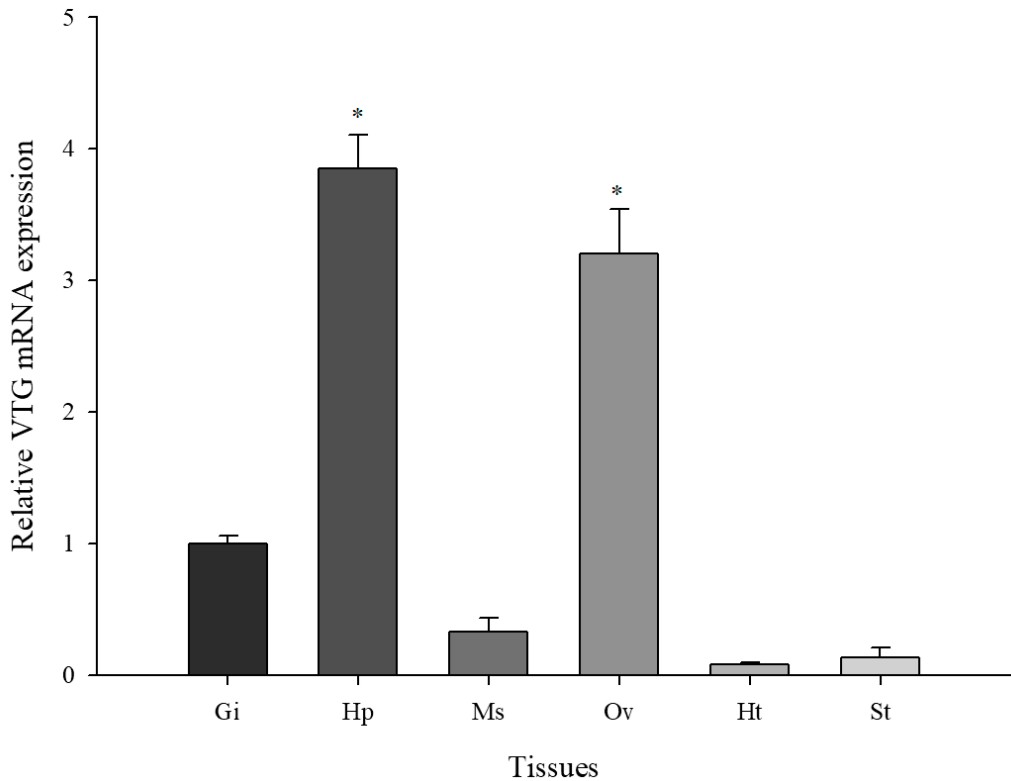

**Figure 3.** Basal expression of *M. japonicus* VTG transcripts among different tissues (Gi, gill; Hp, hepatopancreas; Ms, muscle; Ov, ovaries; Ht, heart; St, stomach). Each tissue was collected from 10 mud crabs. The experiments were performed in triplicate. The data are represented as the mean $\pm$ SD. The relative expression levels of each gene in each tissue were compared with the respective gene expression levels in gills (relative to set value of gill = 1). Significant differences at * $p < 0.05$ are indicated with an asterisk.

### 3.3. VTG Gene Expressions in M. Japonicus Ovaries and Hepatopancreas after DEHP, BPA, or Irgarol Exposures

After an exposure of 24 h, VTG mRNA expression generally increased in *M. japonicus* hepatopancreas exposed to DEHP, BPA or irgarol (Figure 4A). In particular, exposure to 1 and 30 $\mu$g L$^{-1}$ DEHP significantly induced VTG gene expression ($p < 0.05$) relative to the control (when the value of the control = 1). In the ovaries, transcriptional levels of VTG increased slightly after BPA exposure (Figure 4B). In contrast, irgarol slightly decreased VTG expression at a relatively low concentration of 1 $\mu$g L$^{-1}$, although VTG response was somewhat induced at a relatively high irgarol concentration of 30 $\mu$g L$^{-1}$. After exposure to different concentrations of DEHP, VTG gene expression was significantly upregulated at 10 (2.8-fold) and 30 $\mu$g L$^{-1}$ (3.3-fold) of DEHP in a concentration-dependent manner ($p < 0.05$).

The VTG gene response increased in hepatopancreas exposed to DEHP and BPA for 96 h, whereas a decrease in VTG expression resulted from irgarol exposure (Figure 5A). The up-regulation of VTG gene expression was significantly different at a relatively high concentration of BPA (30 $\mu$g L$^{-1}$) ($p < 0.05$). In addition, DEHP exposure significantly induced the up-regulation of the VTG expression in the hepatopancreas. The highest VTG expression was observed at 30 $\mu$g L$^{-1}$ (4.9-fold) DEHP ($p < 0.01$). In contrast, after 96 h of irgarol exposure, VTG decreased in a dose-dependent manner. In the *M. japonicus* ovaries, BPA and DEHP exposure significantly upregulated transcriptional levels of VTG following exposure to all concentrations ($p < 0.05$) (Figure 5B). Similar to the VTG gene response in the hepatopancreas, the VTG expression in the *M. japonicus* ovaries downregulated after irgarol exposure for 96 h. The highest level of VTG expression was found at 30 $\mu$g L$^{-1}$ (5.1-fold) of DEHP ($p < 0.01$).

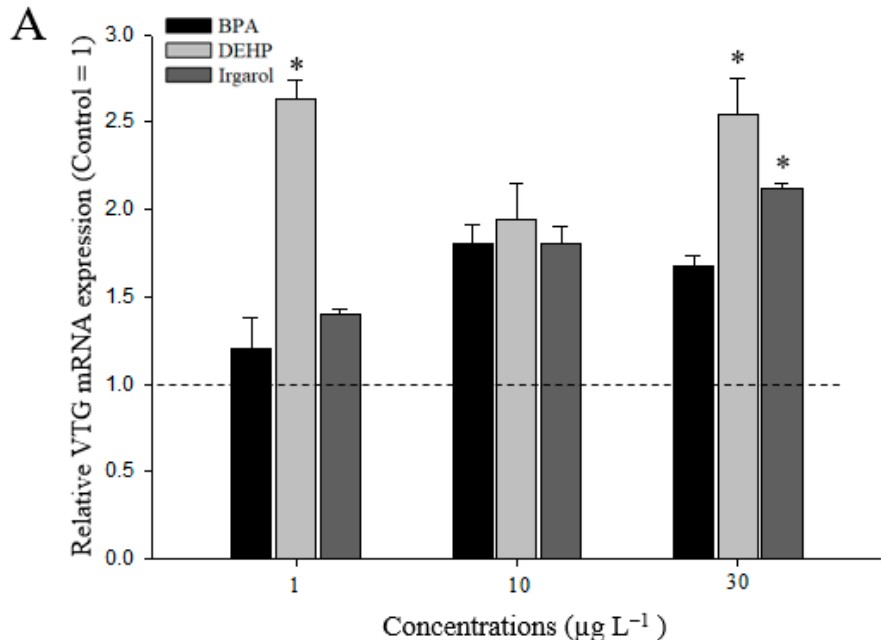

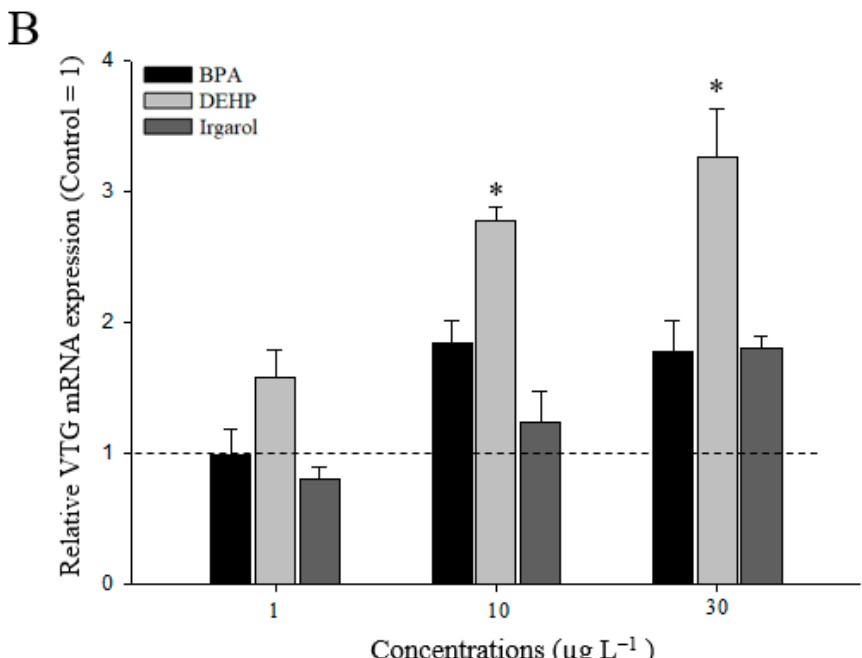

**Figure 4.** Transcriptional expression of VTG gene in *M. japonicus* hepatopancreas (**A**) and ovaries (**B**) exposed to each dose of bisphenol A (BPA), di(2-ethylhexyl) phthalate (DEHP), and irgarol for 24 h. The values were normalized against glyceraldehyde-3-phosphate dehydrogenase (GAPDH). Values of each bar represent the mean ± SD. A statistically significant difference is presented by an asterisk at * $p < 0.05$ as compare to the control (relative control value of VTG = 1).

At 7 d of exposure, there were different responses of the VTG gene between the hepatopancreas and ovary tissues (Figure 6). In the hepatopancreas, significant up-regulation of the VTG gene was observed at relatively low concentrations of 1 µg L$^{-1}$ of BPA (4.3 fold) and DEHP (5.1 fold) (Figure 6A). The responses of VTG gene expression decreased in a concentration-dependent manner. After irgarol exposure, VTG mRNA expression slightly increased in the *M. japonicus* hepatopancreas, compared to the control level, for 7 d. In terms of VTG response in the ovaries, VTG expression was upregulated at all concentrations of BPA, DEHP, or irgarol exposure (Figure 6B). BPA exposure

significantly increased VTG gene expression at 1 (4.3-fold), 10 (4.0-fold), and 30 $\mu g\ L^{-1}$ (5.1-fold) of BPA ($p < 0.05$). DEHP treatment also significantly induced the VTG mRNA level at 1 (4.2 fold), 10 (5.3 fold) and 30 $\mu g\ L^{-1}$ (7.2 fold) of DEHP ($p < 0.01$). The trend of VTG in response to DEHP exposure increased in a dose-dependent manner. In addition, significant expression of VTG was found at a relatively high concentration of 30 $\mu g\ L^{-1}$ (3.8 fold) of irgarol, although VTG gene expression generally increased in *M. japonicus* ovaries exposed to irgarol for 7 d. Furthermore, there are no significant difference in weight changes between the control and treated groups after 7 d exposures to each chemical (Supplementary Figure S1).

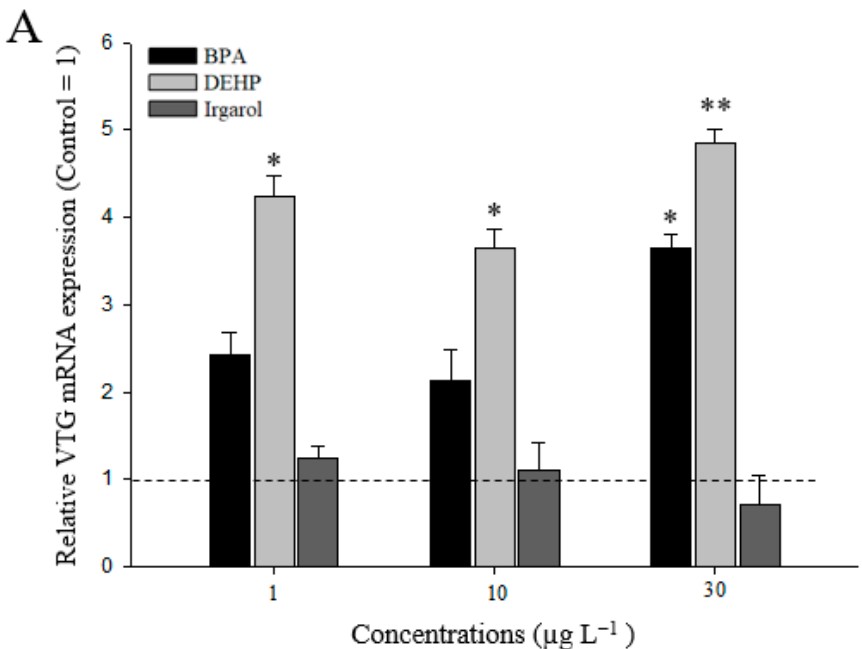

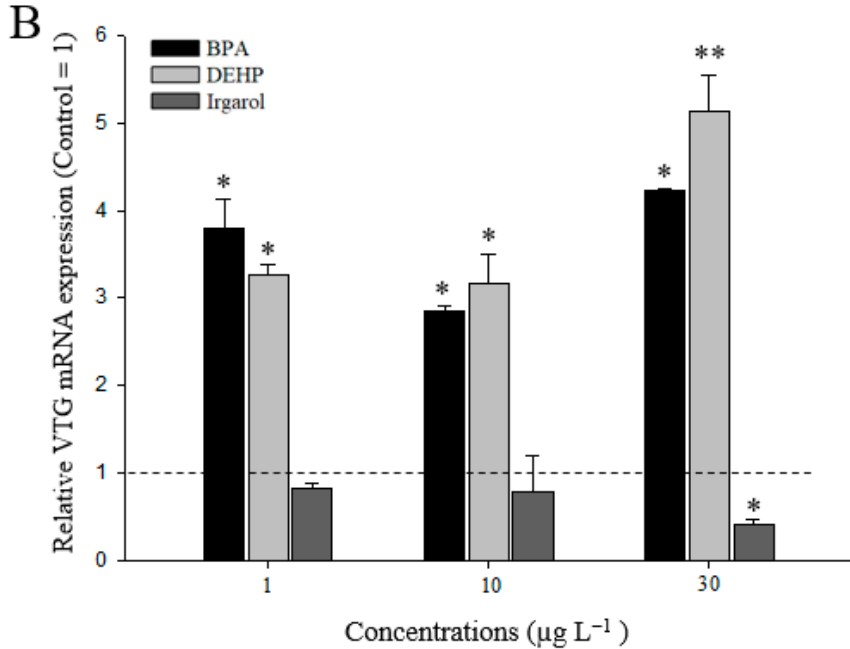

**Figure 5.** Expression of VTG mRNA in *M. japonicus* hepatopancreas (**A**) and ovaries (**B**) after exposure to each dose of BPA, DEHP, and irgarol for 96 h. The values were normalized against GAPDH. Values of each bar represent the mean ± SD. A statistically significant difference is presented by an asterisk at * $p < 0.05$ and ** $p < 0.01$ as compare to the control (relative control value of VTG = 1).

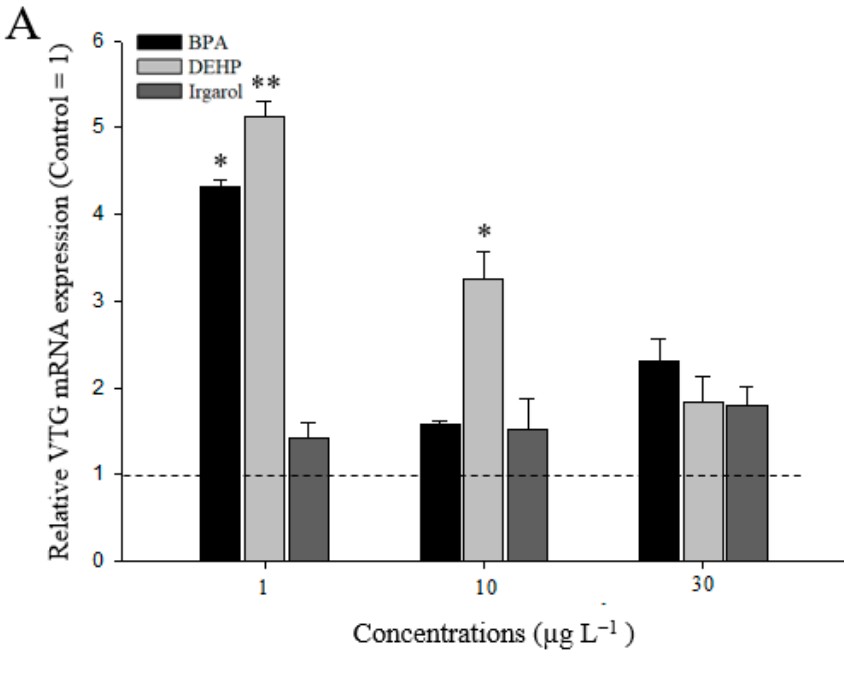

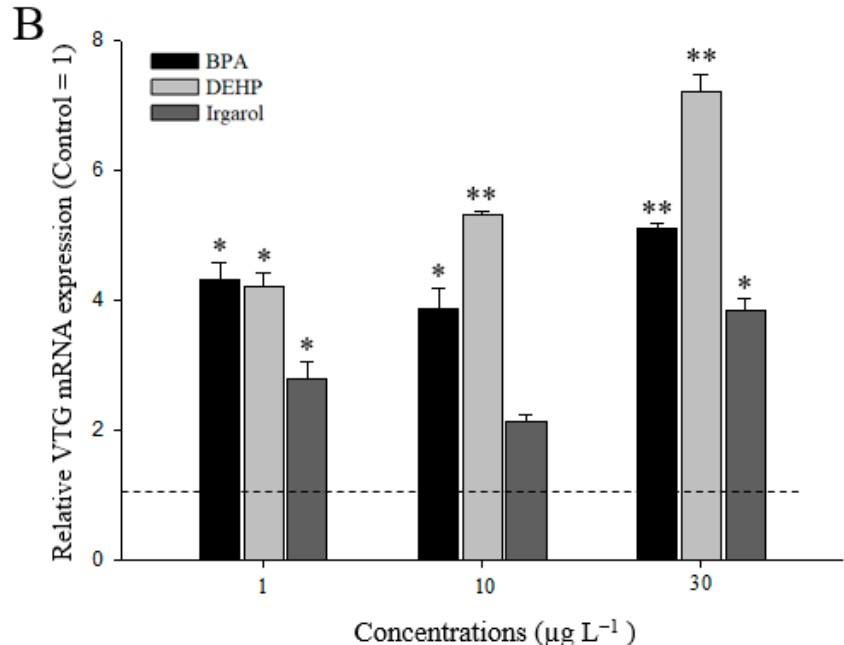

**Figure 6.** Expression of VTG mRNA in *M. japonicus* hepatopancreas (**A**) and ovaries (**B**) after exposure to each dose of BPA, DEHP, and irgarol for 7 d. The values were normalized against GAPDH. Values of each bar represent the mean $\pm$ SD. A statistically significant difference was presented by an asterisk at * $p < 0.05$ and ** $p < 0.01$ as compare to the control (relative control value of VTG = 1).

With different exposure times, VTG gene expression for 24 or 96 h did not differ significantly between the hepatopancreas and ovary tissues ($p > 0.05$). VTG expression levels for 7 d were significantly different between the hepatopancreas and gill ($p < 0.001$). An increase of the VTG gene expression was observed to be time-dependent in *M. japonicus* hepatopancreas exposed to the relatively low concentrations of 1 µg L$^{-1}$ BPA and DEHP (Supplementary Figure S2). In ovary tissue, the up-regulations of VTG gene were found to be time-dependent at all concentrations of BPA and DEHP (Supplementary Figure S3).

## 4. Discussion

To understand VTG gene responses following exposure to environmental chemical pollutants, the cDNA sequence of *M. japonicus* VTG was identified and analyzed using phylogenetic analysis. The three functional domains of the crustacean VTG gene are the lipoprotein domain at the N-terminus (LPD_N), the DUF1943 domain, and the von Willebrand factors type-D domain (VWD) [2,23,33]. In this study, we characterized the *M. japonicus* VTG with 3700 bp ORF, which encodes a protein of 1202 amino acids. The deduced protein region of the *M. japonicus* VTG was predicted as the LPD_N of VTG, which is a conserved region found in various lipid transport proteins [34]. In multiple alignments of the amino acid sequences, *M. japonicus* VTG showed high homology with the protein from *E. sinensis*. The phylogenetic analysis revealed two clusters representing shrimps or crabs among the crustaceans. In these clusters, three clades were represented by crabs, including *M. japonicus*, other crabs, and *H. americanus*.

In this study, it was observed that *M. japonicus* mainly synthesizes VTG in the hepatopancreas and ovaries. In crustaceans, the hepatopancreas is a critical storage system of nutrients and energy in the forms of carbohydrates, proteins, and lipids, which are essential in regulating an energy balance during the reproductive cycle [35]. The hepatopancreas was also reported to be the main source of VTG activity in decapod species, although VTG expression was also found in the ovaries [5]. The primary expression site of the VTG gene in crustaceans has been a controversial topic. VTG production was only observed in the hepatopancreas of *Macrobrachium rosenbergii* (freshwater prawn), *Cherax quadricarinatus* (red claw crayfish), *Pandalus hypsinotus* (coonstriped shrimp) and *P. trituberculatus* [2,36–38]. In contrast, the VTG gene was expressed in both the hepatopancreas and ovaries of *Upogebia major* (mud shrimp) and marine crabs such as *S. paramamosain*, *Carcinus maenas* (green crab), *C. sapidus*, *C. japonica* and *E. sinensis* [9,23,39–41]. We identified that the major sites for VTG synthesis in *M. japonicus* are the hepatopancreas and ovaries.

The endocrine disruptor BPA is presently considered a ubiquitous emerging pollutant in aquatic environments. In this study, exposures to BPA triggered up-regulation of *M. japonicus* VTG expression in the hepatopancreas and ovary tissues for different exposure periods. BPA exposure increased VTG levels in various fish, such as fathead minnows, goldfish, and bluegills [42]. BPA toxicity at lower concentrations ($<15 \ \mu g \ L^{-1}$) showed induced ovarian expression of a steroidogenic gene involving VTG in *G. rarus* [43]. In addition, different modulations of VTG were reported in two VTG genes in the gilthead bream *Sparus aurata* and *G. rarus* [43,44]. These results indicate that induction of VTG following BPA exposure is a major response to the stressful effects of endocrine-disruption.

The endocrine effects of DEHP, a distinctive EDC, are noticeable in marine ecosystems where high concentrations of DEHP have been found in river estuaries and coastal areas [45]. In our study, DEHP exposure generally increased *M. japonicus* VTG expression in the tested tissues. The up-regulation of VTG gene expression in *M. japonicus* was more sensitive to DEHP exposure than to exposure to the other chemicals tested. In addition, transcriptional responses of the VTG gene in the hepatopancreas were found to be significant under relatively low concentrations of DEHP for the different exposure times, whereas VTG was expressed in the ovaries under all concentrations of DEHP tested. Adeogun reported that DEHP exposure induced time- and concentration-dependent increases of VTG levels in *C. gariepinus* [17]. Long exposure to low doses of DEHP reduced oocyte maturation [46] and increased the levels of hepatic VTG transcripts in *G. rarus* [16]. In addition, VTG levels in marine medaka livers increased after DEHP exposure [47]. These results strongly emphasize the suitability of using the VTG gene as a biomarker to assess the environmental presence of DEHP.

Since the use of irgarol as an algaecide has increased, irgarol has been detected frequently in coastal environments [48]. However, there is not enough information about the potential effects of irgarol exposure on the endocrine system of crustaceans. In our study, the responses of VTG gene expression following exposure to irgarol were different. After a 24 h exposure to irgarol, the *M. japonicus* VTG gene was expressed in a dose-dependent manner. The up-regulation of VTG relative to the control decreased in the hepatopancreas and ovaries when exposed to irgarol for 96 h. VTG gene expression recovered

after 7 d of irgarol exposure. These results are the first report of the endocrine effects of irgarol exposure in crustaceans. A recent study reported that irgarol exposure induced changes in thyroid endpoints in the inland silverside fish (*Menidia beryllina*) under salinity and temperature changes [28]. In *A. tenuis*, exposure to irgarol for 7 d resulted in body color changes and reduced transcriptional expression of the HSP90 gene [29]. In a previous study, irgarol exposure induced expression changes of digestion related genes and chitinase genes [17]. Thus, VTG gene expression was affected by environmental chemical pollutants such as BPA, DEHP, and irgarol. After EDC exposures, the expression pattern of the *M. japonicus* VTG gene was correlated with short-term EDC exposure in the hepatopancreas ($p$ = 0.791) and long-term EDC exposure in ovary tissue ($p$ = 0.927).

In conclusion, our results demonstrate that *M. japonicus* is sensitive to several chemical pollutants, as measured by changes of the VTG gene expression in the hepatopancreas and ovaries. The results suggest that the molecular balance of VTG expression levels of intertidal mud crabs appear to be affected by EDCs such as BPA, DEHP, and irgarol at even relatively low concentrations of 1 μg L$^{-1}$. The expression levels of the VTG gene are important factors to be considered for evaluating xenoestrogenic responses. The change of the VTG responses are related with disturbed processes of reproductive physiology such as egg yolk generation in *M. japonicus*, because of the up-regulation of VTG expression in the ovaries after EDC exposures. The crab hepatopancreas is an organ for VTG synthesis and storage of energy accumulation, which can support lipid transportation for oogenesis [49]. EDC exposure also impaired the molecular balance of the VTG level through the disturbance of energy metabolisms in *M. japonicus* hepatopancreas. Up or down-regulation of VTG gene may result in sensitive defense activities of *M. japonicus* crabs to potential EDCs pollutants, given the major function of VTG in stabilizing the endocrine process and triggering lipid uptake for energy storage during crustacean embryogenesis. Our findings suggest that *M. japonicus* VTG may be a useful key to estimate the endocrine effects of antifoulants such as irgarol, BPA, and DEHP. In further studies, we will need to demonstrate the effects of one or more combined exposures of these EDCs in VTG protein synthesis.

**Supplementary Materials:** The following are available online at http://www.mdpi.com/2076-3417/9/7/1401/s1, Figure S1: The relative weight (g) in *M. japonicus* crabs exposed to each dose of BPA, DEHP, and irgarol for 7 d. Figure S2: Transcriptional expression of VTG gene in *M. japonicus* hepatopancreas exposed to each dose of BPA, DEHP, and irgarol for each exposure period. Figure S3: Transcriptional expression of VTG gene in *M. japonicus* ovary exposed to each dose of BPA, DEHP, and irgarol for each exposure period.

**Author Contributions:** Conceptualization: K.P. and I.-S.K.; Methodology: K.P and I.-S.K.; Formal Analysis: K.P., H.J. and D.-K.K.; Investigation: K.P., D.-K.K. and H.J.; Resources: K.P. and I.-S.K.; Writing—Original Draft Preparation: K.P.; Writing—Review and Editing: H.J., D.-K.K. and I.-S.K.; Supervision: I.-S.K.; Project Administration: I.-S.K.; Funding Acquisition: I.-S.K.

**Funding:** This study was supported by the National Research Foundation of Korea, Korea, which is funded by the Korean Government (NRF-2018-R1A6A1A-03024314).

**Conflicts of Interest:** The authors declare no conflict of interest.

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
