# Peer review of "Environmental Pollutants Impair Transcriptional Regulation of the Vitellogenin Gene in the Burrowing Mud Crab (Macrophthalmus Japonicus)"

_applsci, doi:10.3390/app9071401_

Round 1
Reviewer 1 Report
Please correct the spelling of the word 'Marker' in the title (appears as Maker).
The objectives of this study are worthy of investigation and the authors have conducted a study that provides some insights into the effects of three environmental pollutants on VTG expression in the mud crab over an exposure time period of 7 days at a range of concentrations.
Major:
The results are well presented and show some significant effects when compared with reference tissue (Gill) or the control gene GAPDH. My only major concern would be the interpretation of these results. Since the authors only show the changes in VTG expression (relative to GAPDH), it is very difficult to understand what the reported magnitudes of these increases/decreases in expression mean since there are no other control genes to compare with. For example, are there any other genes that could be included in the analysis (relative to GAPDH) that show similar or different levels of expression following exposures to these contaminants? Have any other studies in crab monitored the expression of genes that essentially could be included as controls? If so, these would be important to consider so that the changes in VTG expression could then be compared to genes with predictable expression pattern changes. This would allow the reader to understand what an x-fold change in VTG expression might mean relative to others. Simply comparing VTG expression over time at different dose of chemicals only allows the reader to view the changes of this experimental gene without understanding how other, perhaps previously studies genes, might behave. These types of controls are important beyond the GAPDH normalization.
Minor:
Authors should provide more details for why the doses and time-frames used in this study were selected.
Were there any mortalities in their study.
It would be helpful if the time-course data for the VTG expression was shown on the same graph for each tissue (e.g. ovary at 24h, 96h, and 7 days).
The authors should add more context and perhaps future insights into their last statement: "These results suggest that expression levels of the VTG gene are important factors to be considered for evaluation xenostrogenicresponses"
Author Response
24th March 2019
Applied Science Editor
We are sending the revised manuscript titled “Environmental pollutants impair transcriptional regulation of the vitellogenin gene in the burrowing mud crab (Macrophthalmus japonicus)” for consideration to publish in Applied Science journal.
Vitellogenesis is a pivotal reproductive process of the yolk formation in crustaceans. Vitellogenin (VTG) is the precursor of main yolk proteins and synthesized by endogenous estrogens. The intertidal mud crab (Macrophthalmus japonicus) inhabits sediment and is a good indicator for assessing polluted benthic environments. The purpose of this study was to identify potential responses of M. japonicus VTG under environmental stresses caused by chemical pollutants, such as 1, 10 and 30 µg L-1 concentrations in di(2-ethylhexyl) phthalate (DEHP), bisphenol A (BPA) and irgarol. We characterized the M. japonicus VTG gene and analyzed the transcriptional expression of VTG mRNA in M. japonicus exposed to various chemicals and exposure periods. A phylogenetic analysis revealed that the M. japonicus VTG clustered closely with Eriocheir sinensis (Chinese mitten crab) VTG, in contrast with another clade that included the VTG ortholog of other crabs. The basal level of VTG expression was the highest in the hepatopancreas and ovary and tissues. VTG expression significantly increased in the ovary and hepatopancreas after 24 h exposure to DEHP. Increased responses of VTG transcripts were found in M. japonicus exposed to DEHP and BPA for 96 h; however, VTG expression decreased in both tissues after irgarol exposure. After an exposure of 7 d, VTG expression significantly increased in the ovary and hepatopancreas for all concentrations of all chemicals. These results suggest that crustacean embryogenesis and endocrine process are impaired by the environmental chemical pollutants DEHP, BPA and irgarol.
We believe that our findings would satisfy your journal requirements for the publication of this paper. Therefore, please be kind enough to evaluate our manuscript and inform us your comments.
The English in the manuscript has been corrected by a professional science-editing services (Haricos). In addition, our specific responses to the points raised by the reviewers are presented below.
Reviewer #1_applsci-455253
The objectives of this study are worthy of investigation and the authors have conducted a study that provides some insights into the effects of three environmental pollutants on VTG expression in the mud crab over an exposure time period of 7 days at a range of concentrations.
Major:
The results are well presented and show some significant effects when compared with reference tissue (Gill) or the control gene GAPDH. My only major concern would be the interpretation of these results. Since the authors only show the changes in VTG expression (relative to GAPDH), it is very difficult to understand what the reported magnitudes of these increases/decreases in expression mean since there are no other control genes to compare with. For example, are there any other genes that could be included in the analysis (relative to GAPDH) that show similar or different levels of expression following exposures to these contaminants? Have any other studies in crab monitored the expression of genes that essentially could be included as controls? If so, these would be important to consider so that the changes in VTG expression could then be compared to genes with predictable expression pattern changes. This would allow the reader to understand what an x-fold change in VTG expression might mean relative to others. Simply comparing VTG expression over time at different dose of chemicals only allows the reader to view the changes of this experimental gene without understanding how other, perhaps previously studies genes, might behave. These types of controls are important beyond the GAPDH normalization.
à We evaluated the best reference gene for qRT-PCR on M. japonicus crab. Evaluation of the best reference gene for qPCR is essential. GAPDH, b-actin, and elongation factor a1(Efa) were analyzed to find the best house-keeping gene for the exposure experiment. Expression stability was analyzed using geNorm ranking and the comparative delta-Ct method as described (Silver et al., 2006 and Vandesompele et al., 2002). Among these, the expression of GAPDH was identified as the most stable (data not shown), hence the qRT-PCR data was normalized with GAPDH.
Minor:
Authors should provide more details for why the doses and time-frames used in this study were selected.
à In this study, the nominal concentrations for EDCs were 1, 10 and 30 µg L -1. Due to its widespread use, BPA is reported to be present in the aquatic ecosystem in concentrations between 0.0005 and 12 mg L-1 BPA (Balbi et al., 2016). In addition, BPA has been observed at relatively high concentrations (up to 21 μg L-1 ) in rivers and lakes: 12 μg L-1 in some polluted river water in the USA and 3.92 μg L-1 in the Pearl River estuary in China (Kolpin et al., 2002; Crain et al., 2007; Dong et al., 2009). DEHP was reported to reach a level of 45.73 μg g−1 in the Yellow River, and 1.56 μg g−1 (dry weight) in Qiantang river sediment in China (Sha et al., 2007; Sun et al., 2013). We selected exposure concentrations based on relative environmental concentrations and acute toxicity test in previous results (Park et al., 2016; Park et al., 2019).
Were there any mortalities in their study.
à In the previous study, we evaluated survival rate of M. japonicus crab after ECDs (BPA, DEHP, Irgarol) exposures (Park et al., 2016; Park et al., 2019). In irgarol exposure, M. japonicus exposed to 30 μg L-1 began to die on day 1 (92.3% survival), and survival declined on day 2 (84.6% survival) and day 4 (76.9% survival); it continued to decline until day 7, ending with a 69.2% cumulative survival rate. In BPA exposure, crabs exposed to 30 μg L−1 BPA started to die at day 1 (91.9%), and continually declined until day 7 (64.9%), and the survival rate was lower than 1 with 10 μg L−1 BPA at day 7 (73.7%, 92.1%). In DEHP exposure, in 30 μg L−1 DEHP, M. japonicus started to die at day 1 (70.0%), and continually declined until day 7 (25.0%). The detailed information for mortalities was described in previous studies (Park et al., 2016; Park et al., 2019).
It would be helpful if the time-course data for the VTG expression was shown on the same graph for each tissue (e.g. ovary at 24h, 96h, and 7 days).
à We added the time-course data for the VTG gene expression in supplementary Fig. 1 and Fig. 2 of the revised manuscript. Related content is included in the revised manuscript (lines 239~243 on page 11).
The authors should add more context and perhaps future insights into their last statement: "These results suggest that expression levels of the VTG gene are important factors to be considered for evaluation xenostrogenic responses"
à We added the new paragraph into last part of the discussion section in the revised manuscript (lines 311~328 on pages 14~15): “ In conclusion, our results demonstrated that M. japonicus is sensitive to several chemical pollutants as measured by changes of the VTG gene expression in the hepatopancreas and ovary. The results suggest that intertidal mud crab appear to be affected on a molecular balance of VTG expression level by EDCs such as BPA, DEHP and irgarol at even relatively low concentrations of 1 µg L-1. The expression levels of the VTG gene are important factors to be considered for evaluating xenoestrogenic responses. The change of the VTG responses are related with disturbed process of reproductive physiology such as egg yolk generation in M. japonicus, because of up-regulation of VTG expression in ovary after EDCs exposures. Crab hepatopancreas is an organ for VTG synthesis and storage of energy accumulation, which can support lipid transportation for oogenesis [52]. The EDCs exposure also impaired molecular balance of VTG level through disturbance of energy metabolisms in M. japonicus hepatopancreas. Up or down-regulation of VTG gene may result in sensitive defense activities of M. japonicus crabs to potential EDCs pollutants, given the major function of VTG in stabilizing endocrine process and triggering lipid uptake for energy storage during crustacean embryogenesis. Our findings suggest that M. japonicus VTG may be a useful key to estimate the endocrine effects of antifoulant, such as irgarol, as well as BPA and DEHP. In further study, we will need to demonstrate the effects of one or combination exposures of these EDCs in VTG protein synthesis.”
Thanks in advance.
Yours sincerely,
Ihn-Sil Kwak, Ph.D. Preofessor
Dept. of Environmental Oceanography, Chonnam National University,
Chonnam, 550-749, Korea.
E-mail address: iskwak@chonnam.ac.kr

Reviewer 2 Report
Park et. al. demonstrate that M. japonicus is sensitive to several chemical pollutants as measured by increased expression of the VTG gene in the hepatopancreas and ovary. The results presented are interesting, however only show changes in RNA expression. While RNA expression often correlates with protein expression, confirmation of changes in protein expression, at least at key time points, would confirm the significance of the data presented. Additionally, it could be expected that many of these pollutants are not found in isolation. It would be interesting to know if combinations of these pollutants have more than additive effects (ie irgarol + DEHP or BPA).
Author Response
24th March 2019
Applied Science Editor
We are sending the revised manuscript titled “Environmental pollutants impair transcriptional regulation of the vitellogenin gene in the burrowing mud crab (Macrophthalmus japonicus)” for consideration to publish in Applied Science journal.
Vitellogenesis is a pivotal reproductive process of the yolk formation in crustaceans. Vitellogenin (VTG) is the precursor of main yolk proteins and synthesized by endogenous estrogens. The intertidal mud crab (Macrophthalmus japonicus) inhabits sediment and is a good indicator for assessing polluted benthic environments. The purpose of this study was to identify potential responses of M. japonicus VTG under environmental stresses caused by chemical pollutants, such as 1, 10 and 30 µg L-1 concentrations in di(2-ethylhexyl) phthalate (DEHP), bisphenol A (BPA) and irgarol. We characterized the M. japonicus VTG gene and analyzed the transcriptional expression of VTG mRNA in M. japonicus exposed to various chemicals and exposure periods. A phylogenetic analysis revealed that the M. japonicus VTG clustered closely with Eriocheir sinensis (Chinese mitten crab) VTG, in contrast with another clade that included the VTG ortholog of other crabs. The basal level of VTG expression was the highest in the hepatopancreas and ovary and tissues. VTG expression significantly increased in the ovary and hepatopancreas after 24 h exposure to DEHP. Increased responses of VTG transcripts were found in M. japonicus exposed to DEHP and BPA for 96 h; however, VTG expression decreased in both tissues after irgarol exposure. After an exposure of 7 d, VTG expression significantly increased in the ovary and hepatopancreas for all concentrations of all chemicals. These results suggest that crustacean embryogenesis and endocrine process are impaired by the environmental chemical pollutants DEHP, BPA and irgarol.
We believe that our findings would satisfy your journal requirements for the publication of this paper. Therefore, please be kind enough to evaluate our manuscript and inform us your comments.
The English in the manuscript has been corrected by a professional science-editing services (Haricos). In addition, our specific responses to the points raised by the reviewers are presented below.
Reviewer #2_applsci-455253
Comments and Suggestions for Authors
Park et. al. demonstrate that M. japonicus is sensitive to several chemical pollutants as measured by increased expression of the VTG gene in the hepatopancreas and ovary. The results presented are interesting, however only show changes in RNA expression. While RNA expression often correlates with protein expression, confirmation of changes in protein expression, at least at key time points, would confirm the significance of the data presented. Additionally, it could be expected that many of these pollutants are not found in isolation. It would be interesting to know if combinations of these pollutants have more than additive effects (ie irgarol + DEHP or BPA).
à In the present study, we analyzed the changes in VTG expression levels to compare with three control genes (GAPDH, b-actin, and elongation factor a1). This would allow the reader to understand what an x-fold change in VTG expression might mean relative response by pollutant exposures. In further study, we will need to demonstrate the effects of one or combination exposures of these pollutants in VTG protein expression (lines 327~328 on page 15).
Thanks in advance.
Yours sincerely,
Ihn-Sil Kwak, Ph.D. Preofessor
Dept. of Environmental Oceanography, Chonnam National University,
Chonnam, 550-749, Korea.
E-mail address: iskwak@chonnam.ac.kr

Reviewer 3 Report
In this study, Park et al. have analyzed the Vitellogenin (VTG) mRNA levels as a marker for endocrine disrupting chemicals (such as bisphenol A (BPA), di(2-ethylhexyl) phthalate (DEHP) and irgarol) exposure in the intertidal mud crab (Macrophthalmus japonicus). The authors observed that the VTG transcript levels are higher in ovary and hepatopancreas at a basal level. Further, it was increased at 24h and 96h in BPA and DEHP exposed group but irgarol showed the opposite trend. At the end of the study exposure of 7 days, VTG mRNA level statistically increased in all the concentration of all chemicals. Authors suggest the conclusion based on the observations and results that crustacean embryogenesis and endocrine process are impaired by the environmental endocrine disruptors (DEHP, BPA and irgaro).
Though the paper is well written, there are some concerns.
Comments:
1. The title may be changed since only VTG mRNA only analyzed in the study the word “potential” in the title seems odd; the title must be changed according to the observation.
2. Abstract: Include the concentrations of endocrine disrupting chemicals may be beneficial for readers.
3. Introduction: Logical link missing between paragraphs and firm hypothesis should be mentioned instead of objectives.
4. Materials methods: Exposure experiments: What is the time interval of seawater change in each tank; whether treatment chemicals also changed for >24h experiments? Details must be included.
5. Is there any mortality in animals?
6. What is the biological and technical replicate for the experiments? (Did 10 animals per group used for RT-PCR?)
7. Provide weight change in the animals from basal point to 7days.
8. Statistical Analysis: though it was mentioned p<0.01(**) it was missing in figure legends and certain bar diagrams seem statistical significant but significant (*,**) notations missing. Do reanalysis whether it reaches statistical significance.
9. Why Gill VTG mRNA levels set as a baseline? What is the rationale?
10. Some part of the discussion is not clear it was too descriptive. Those sentences and paragraphs need to rewrite.
11. The manuscript should be checked for grammatical corrections.
The paper can be accepted after minor revision.
Author Response
24th March 2019
Applied Science Editor
We are sending the revised manuscript titled “Environmental pollutants impair transcriptional regulation of the vitellogenin gene in the burrowing mud crab (Macrophthalmus japonicus)” for consideration to publish in Applied Science journal.
Vitellogenesis is a pivotal reproductive process of the yolk formation in crustaceans. Vitellogenin (VTG) is the precursor of main yolk proteins and synthesized by endogenous estrogens. The intertidal mud crab (Macrophthalmus japonicus) inhabits sediment and is a good indicator for assessing polluted benthic environments. The purpose of this study was to identify potential responses of M. japonicus VTG under environmental stresses caused by chemical pollutants, such as 1, 10 and 30 µg L-1 concentrations in di(2-ethylhexyl) phthalate (DEHP), bisphenol A (BPA) and irgarol. We characterized the M. japonicus VTG gene and analyzed the transcriptional expression of VTG mRNA in M. japonicus exposed to various chemicals and exposure periods. A phylogenetic analysis revealed that the M. japonicus VTG clustered closely with Eriocheir sinensis (Chinese mitten crab) VTG, in contrast with another clade that included the VTG ortholog of other crabs. The basal level of VTG expression was the highest in the hepatopancreas and ovary and tissues. VTG expression significantly increased in the ovary and hepatopancreas after 24 h exposure to DEHP. Increased responses of VTG transcripts were found in M. japonicus exposed to DEHP and BPA for 96 h; however, VTG expression decreased in both tissues after irgarol exposure. After an exposure of 7 d, VTG expression significantly increased in the ovary and hepatopancreas for all concentrations of all chemicals. These results suggest that crustacean embryogenesis and endocrine process are impaired by the environmental chemical pollutants DEHP, BPA and irgarol.
We believe that our findings would satisfy your journal requirements for the publication of this paper. Therefore, please be kind enough to evaluate our manuscript and inform us your comments.
The English in the manuscript has been corrected by a professional science-editing services (Haricos). In addition, our specific responses to the points raised by the reviewers are presented below.
Reviewer #3_applsci-455253
In this study, Park et al. have analyzed the Vitellogenin (VTG) mRNA levels as a marker for endocrine disrupting chemicals (such as bisphenol A (BPA), di(2-ethylhexyl) phthalate (DEHP) and irgarol) exposure in the intertidal mud crab (Macrophthalmus japonicus). The authors observed that the VTG transcript levels are higher in ovary and hepatopancreas at a basal level. Further, it was increased at 24h and 96h in BPA and DEHP exposed group but irgarol showed the opposite trend. At the end of the study exposure of 7 days, VTG mRNA level statistically increased in all the concentration of all chemicals. Authors suggest the conclusion based on the observations and results that crustacean embryogenesis and endocrine process are impaired by the environmental endocrine disruptors (DEHP, BPA and irgaro).
Though the paper is well written, there are some concerns.
Comments:
1.The title may be changed since only VTG mRNA only analyzed in the study the word “potential” in the title seems odd; the title must be changed according to the observation.
à We changed the title as “Environmental pollutants impair transcriptional regulation of the vitellogenin gene in the burrowing mud crab (Macrophthalmus japonicus)” in the revised manuscript.
2. Abstract: Include the concentrations of endocrine disrupting chemicals may be beneficial for readers.
à Tested concentrations of EDCs were included in abstract section of the revised manuscript
3. Introduction: Logical link missing between paragraphs and firm hypothesis should be mentioned instead of objectives.
à We revised the whole introduction section of the revised manuscript following logical linking content (lines 55~58, lines 75~79, lines 87~89 and switch of the second and third paragraphs).
4. Materials methods: Exposure experiments: What is the time interval of seawater change in each tank; whether treatment chemicals also changed for >24h experiments? Details must be included.
à We described the detailed materials and methods in the revised manuscript (lines 116~118 on page 5) as the following sentences “The seawater was changed every day by adding the equivalent concentration of each chemical during the experiment. No food was provided during the experimental period.”
5. Is there any mortality in animals?
à In the previous study, we evaluated survival rate of M. japonicus crab after ECDs (BPA, DEHP, Irgarol) exposures (Park et al., 2016; Park et al., 2019). In irgarol exposure, M. japonicus exposed to 30 μg L-1 began to die on day 1 (92.3% survival), and survival declined on day 2 (84.6% survival) and day 4 (76.9% survival); it continued to decline until day 7, ending with a 69.2% cumulative survival rate. In BPA exposure, crabs exposed to 30 μg L−1 BPA started to die at day 1 (91.9%), and continually declined until day 7 (64.9%), and the survival rate was lower than 1 with 10 μg L−1 BPA at day 7 (73.7%, 92.1%). In DEHP exposure, in 30 μg L−1 DEHP, M. japonicus started to die at day 1 (70.0%), and continually declined until day 7 (25.0%). The detailed information for mortalities was described in previous studies (Park et al., 2016; Park et al., 2019).
6. What is the biological and technical replicate for the experiments? (Did 10 animals per group used for RT-PCR?)
à For each chemical exposure, the crabs (n = 120 for each chemical exposure) were randomly divided into four experimental groups (corresponding to the 1, 10 and 30 µg L-1 treatment solutions, as well as solvent controls). 10 crabs were treated with one of three concentrations of BPA, DEHP or irgarol. Three animals were subjected to tissue extraction for each time intervals in each chemical exposure conditions as well as control crabs. All experiments were triplicates with independent samples. Thus, after each exposure experiment, total 9 animals per group at each time were used for gene expression analysis using RT-PCR. These content included in Materials and method section (lines 115~116 on page 5).
7. Provide weight change in the animals from basal point to 7days.
à There is no significant difference of crab weight between control and treated groups, although the weight of M. japonicus crab was slightly more decreased in treated groups than control groups. The related data was included in Supplementary Fig. 3 and in the revised manuscript (lines 234~236 on page 11).
8. Statistical Analysis: though it was mentioned p<0.01(**) it was missing in figure legendsand certain bar diagrams seem statistical significant but significant (*,**) notations missing. Do reanalysis whether it reaches statistical significance.
à We revised the legends of Fig. 5A and Fig. 6B as *P < 0.05 and **P < 0.01 in the revised manuscript. We also revised statistical significances on the bar diagrams of the Fig. 4 and Fig. 6 after statistical reanalysis.
9. Why Gill VTG mRNA levels set as a baseline? What is the rationale?
à Gill is involved in many cellular functions for transcriptional regulation of metabolism, maintaining homeostasis for stress and adaptability and plays a significant role in modulation of ion transport. In contrast to other tissues, the gill tissue interacts first with the environmental changes. For this reasons, gill VTG mRNA levels used as a baseline in tissue distribution analysis of VTG gene. In addition, we used gene expression level in M. japonicus gill as the baseline in the previous studies (Nikapitiya et al., 2014 and 2015).
10. Some part of the discussion is not clear it was too descriptive. Those sentences and paragraphs need to rewrite.
à We revised the discussion section of the revised manuscript (lines 263~270 on page 12, lines 311~328 on pages 14~15).
11. The manuscript should be checked for grammatical corrections.
à The English in the revised manuscript was corrected by a professional science-editing services (Haricos).
The paper can be accepted after minor revision.
Thanks in advance.
Yours sincerely,
Ihn-Sil Kwak, Ph.D. Preofessor
Dept. of Environmental Oceanography, Chonnam National University,
Chonnam, 550-749, Korea.
E-mail address: iskwak@chonnam.ac.kr

Round 2
